# A Unifying Normative Framework of Decision Confidence

**Amelia M. Johnson**
Department of Computer Science
University of Washington
Seattle, WA 98195
ameliamj@uw.edu

**Micheal A. Buice**
Allen Institute
Seattle, WA 98109
michaelbu@alleninstitute.org

**Koosha Khalvati**
Allen Institute
Seattle, WA 98109
koosha.khalvati@alleninstitute.org

## Abstract

Self-assessment of one's choices, i.e., confidence, is the topic of many decision neuroscience studies. Computational models of confidence, however, are limited to specific scenarios such as between choices with the same value. Here we present a normative framework for modeling decision confidence that is generalizable to various tasks and experimental setups. We further drive the implications of our model from both theoretical and experimental points of view. Specifically, we show that our model maps to the planning as an inference framework where the objective function is maximizing the gained reward and information entropy of the policy. Moreover, we validate our model on two different psychophysics experiments and show its superiority over other approaches in explaining subjects' confidence reports [1].

## 1 Introduction

Self-assessment of one's choices, i.e., decision confidence, plays a key role in long-term decision-making and learning [1]. This assessment helps the decision maker improve their model of the outside world and consequently gain higher utility in the future [2, 3]. Due to this critical role, confidence has been the topic of many theoretical and experimental decision neuroscience studies. However, primarily focused on in perceptual decision-making experiments, confidence is mainly mathematically defined only for scenarios where different choices have the same potential reward. In these situations, based on some perceptual cues, the goal is to pick the "correct" choice instead of all other equally "incorrect" ones. Consequently, confidence is defined as the probability of choosing the correct option [1, 4]. In these setups, *perception confidence*, i.e., "what is the probability that my observation was correct?" and *decision confidence*, "what is the probability that I made the correct decision?" are inseparable because the choices do not differ in value.

In the real world, though, different actions lead to various potential rewards, and this variation in reward could influence confidence in decisions, differentiating it from perception confidence. A rustling sound is more likely to be because of the wind rather than a predator approaching. Still, everyone becomes more vigilant when hearing that sound as one of the possibilities is potentially life-threatening despite being unlikely. Moreover, everyone would be confident about the rationality

---

[1]The majority of this research was done while the first author was an intern at the Allen Institute.

of this decision. Notably, most experiments where different choices vary in value, known as value-based decision-making, do not involve uncertainty in perception [5]. These works are mostly about memory retrieval and valuation of different objects and do not study the interaction between value and perceptual uncertainty [6]. Here, we present a normative framework to formally define and assess decision confidence in a general scenario involving uncertainty about the outside world, prior knowledge of the decision maker about the world, and different utility functions for available choices. We model decision confidence as "probability of making the best decision". In mathematical language, decision confidence is the probability of being optimal over a sequence of states and actions given the policy.

We further show that our approach to modeling subjects' confidence in their decision equals to planning as inference framework [7, 8]. This framework maps to a reinforcement learning agent whose objective function is to jointly maximize the reward and the information entropy of the policy (also called maximum entropy reinforcement learning) [9, 10]. Moreover, we validate our framework by testing it on two different experiments on confidence evaluation and explaining its implications [11, 5].

## 2 Modelling Background and Problem Definition

Markov Decision Processes are graphical models used for optimal sequential decision-making in artificial agents. Recently, these frameworks have been applied successfully in modeling the behavior of subjects across various decision-making tasks and species in cognitive neuroscience [12, 13, 14, 15].

### 2.1 Fully and Partially Observable Markov Decision Processes

Formally, a Markov Decision Process (MDP) is a tuple $(S, A, T, R)$ describing a Markovian system where $S$ is the finite set of states, $A$ is the finite set of actions, $T = p(s'|a, s)$ is the transition function between states, and $R$ is a bounded function representing the reward of each action in each state, $r(s, a)$. The goal of an MDP agent is to come up with the recipe of action selection, called *policy*, to maximize its total reward within *horizon* $H$. As the system is Markovian, each policy can be defined as a probability distribution of actions given the state, shown by $\pi(a|s)$. The optimal policy $\pi^*$ is the policy that attains the goal of the agent:

$$\pi^* = \arg\max_{\pi} \mathbb{E}_{(s_t, a_t) \sim \pi(a_t|s_t)} \left[ \sum_{t=1}^{H} r(s, a_t) \right]. \tag{1}$$

The optimal policy can be obtained in polynomial time in the size of the state space, e.g., by using dynamic programming algorithms.

In most real-world situations, however, the environment is only partially observable, making the agent uncertain about the current state of the world. Partially Observable MDP (POMDP) models these situations by adding the concepts of observation, observation function, and the belief state to the MDP framework [16]. POMDP is formally defined as a tuple $(S, A, Z, T, P, R)$ where $S$, $A$, $T$ and $R$ have the same definition as MDP. $Z$ is the finite set of observations. In addition, $P = p(z|s, a)$ is the observation function representing the probability of each observation, given the state and chosen action. A POMDP agent does not know the current state of the environment. Therefore, starting from a prior, called initial belief state $b_1$, it updates the posterior probability distribution over the states with each observation and action:

$$b_t(s) \propto p(z_t|s, a_{t-1}) \sum_{s' \in S} p(s|s', a_{t-1}) b_{t-1}(s'). \qquad (2 \leq t \leq H) \tag{2}$$

In a POMDP, each policy can be represented as a mapping from belief states to a probability distribution over actions, i.e., $\pi(a|b)$ with the optimal policy $\pi^*$ being the mapping that maximizes the expected total reward:

$$\pi^* = \arg\max_{\pi} \mathbb{E}_{(b_t, a_t) \sim \pi(a_t|b_t)} \left[ \sum_{t=1}^{H} \sum_{s \in S} b_t(s) r(s, a_t) \right] = \arg\max_{\pi} \mathbb{E}_{(b_t, a_t) \sim \pi(a_t|b_t)} \left[ \sum_{t=1}^{H} r(b_t, a_t) \right]$$

$$\tag{3}$$

where $r(b_t, a_t) = \sum_{s \in S} b_t(s) r(s_t, a_t)$ is the expected reward of belief state $s_t$ if the agent choose action $a_t$.

POMDP can be viewed as a fully observable MDP with the state space of the belief state space of the original environment. Therefore, the optimal policy can be obtained in polynomial time in the size of the belief state. Since the belief state is a probability distribution over states, its space's size is exponential in the size of the state space. This means that transforming the POMDP to an MDP is not helpful unless the number of states is extremely low (less than 10) or the belief state can always be represented by a distribution of a few parameters. An example of the latter case is Kalman-filter-like environments where the belief state can always be represented with a Gaussian distribution with two parameters of $\mu_t$ and $\sigma_t$.

## 2.2 Models of confidence in Cognitive Neuroscience

One of the applications of POMDPs and similar Bayesian frameworks is modeling the behavior "perceptual decision making" where the subject should select the "correct" choice based on some sensory observations to get reward [17]. As the term "correct" suggests, the reward function is symmetrical among different choices. In some of these studies, subjects also report their confidence in their choice. Numerous experiments have demonstrated that trained subjects perform similarly to an optimal agent such as POMDP. Moreover, their confidence in their choice closely matches the probability of choosing the correct option, i.e., the posterior probability of the most probable choice [1, 18] (or the sum of belief states that leads to the most probable choice). This close match is also called *Bayesian confidence hypothesis* [19]. Notably, due to reward symmetry in these situations, *perception confidence*, modeled as the belief about the hidden state, is inseparable from *decision confidence*, which would be the belief about the decision. More specifically, these experiments cannot flesh out the interactions between value, perception, and confidence and cannot test theoretical models for these interactions.

There are a few confidence experiments with asymmetries in the reward function. Models and methods of these studies, however, are all descriptive/statistical, e.g., positive correlation between confidence report and reward value [20, 21, 11]. Therefore, as opposed to *normative models*, these methods do not explain the reason behind the relationship of confidence with perception, prior, and rewards in a systematic and generalizable manner. Moreover, some experimental and theoretical works have studied "value-based confidence" in the context of "value-based decision making" where the task is to choose an object between offered options [22, 6, 5]. There is no perceptual ambiguity in these studies, and the variance in response is primarily due to the variability of the valuation of objects at different time points based on memory [6].

Finally, note that the optimality assumption in proposed models, such as the Bayesian confidence hypothesis, does not necessarily imply that the subject's confidence and performance are equal on average. As opposed to an AI POMDP agent, the subject is unaware of the exact generative functions of the environment ($S$, $T$, and $P$). The internal world model of the subject is built and learned by training. Therefore, the type and amount of training and feedback and the subject's capability and motivation in creating the accurate internal world model affect confidence. However, given the internal model, the behavior is optimal, e.g., the perception confidence follows the maximum belief state [18, 19]. We will explain this in more detail in our empirical evaluation and discussions.

## 2.3 Problem definition

We aim to formally define and test a normative framework that explains the interaction of perception and reward/value with the subject's confidence about their decision. Before presenting the model, we discuss a thought experiment as the motivation. This experiment consists of a set of trials, each of which begins with the presentation of a Gabor filter tilted left or right. The subject is asked to report their perception of the direction of the Gabor filter (left or right) and their confidence in their decision. Choosing the incorrect direction leads to no reward. Moreover, if correct, the direction left produces a reward 10 times the reward of the direction right. In an example trial, the subject believes the direction is right with a 0.7 probability. Assuming they want to gain a higher reward, they will choose the direction left due to the higher expected value, namely 0.3 versus 0.7. However, their perception confidence or "probability of choosing the correct/rewarding direction" is 0.3. On the other hand, their confidence in their decision, "likelihood of making the best decision" is probably

higher than 0.5 (because they chose it). Notably, while being higher than 0.5, their confidence in this trial is likely lower than in trials where they think the stimulus direction is left. However, from the strict optimality point of view, the decision confidence is 1 in both cases. Therefore, while a strictly optimal framework like POMDP offers a normative model of confidence (by explaining why confidence equals 1 through the lens of optimality), such a model is not aligned with reality.

Our goal is to systematically define and test decision confidence that includes all main aspects of decision-making, including reward, priors, and perceptual cues (observations) from the perspective of an optimal probabilistic decision-maker that is also aligned with reality. Notably, heuristics that combine reward and belief, such as the ratio of expected values of different actions, might work, but they are totally arbitrary and do not generalize to different tasks. We are looking for a *normative model* that explains *why* such a relationship exists between confidence and task parameters in a generalizable fashion.

## 3   Model

To formally model the decision confidence, we use the idea of *optimality* in the POMDP framework, where optimality is a binary variable that reflects receiving the maximum reward. The probability of optimality describes a subject's confidence or internal sense of whether their decision was optimal. This can be viewed as an agent choosing actions to maximize their total reward, modeled by a POMDP, and later evaluating the optimality of their decision.

Notably, the decision-making process might involve multiple actions. Therefore, what we refer to as "decision" is, in fact, a sequence of actions, given an observation after each action $a_1, z_2, a_2, z_3, \ldots, a_H$ which is called trajectory $\tau$. As the agent knows the observation function and the system is Markovian, this trajectory could also be expressed with belief states instead of observations:

$$\tau = a_1, b_2, a_2, b_3, \ldots, a_H \tag{4}$$

Consequently, the probability of observing trajectory $\tau$ in an optimal agent is $p(\tau|O = 1, b_1)$. This probability should be maximum for a trajectory that is generated by the optimal policy $\pi^*$.

With strict interpretation of optimality, only trajectories that are generated from the optimal policy $\pi^*$ are optimal. This means that confidence is 1 for these trajectories and 0 for others. This "hard" definition of optimality punishes all non-optimal trajectories the same, meaning that a trajectory generated by a suboptimal policy is considered as non-optimal as a trajectory with the minimum possible outcome. This is not ideal—especially when the agents are humans, as humans are inherently suboptimal.

Given the fact that the system is Markovian and each policy is a mapping from belief states to actions, the probability of optimality can be expressed for each action given the belief state, i.e., $p(o_t = 1|a_t, b_t)$ (Fig. 1, left plot). With this representation, the probability of optimality for the whole trajectory can be expressed as $p(o_{1:H} = 1|\tau)$. Consequently, the probability of a trajectory being optimal is:

$$p(\tau|o_{1:H}) \propto p(\tau, o_{1:H}) = p(b_0) \prod_{t=0}^{H} p(o_t = 1|b_t, a_t)p(b_{t+1}|b_t, a_t). \tag{5}$$

If the belief state dynamics is deterministic, i.e., each action in each state leads to exactly one state and one observation, we would have:

$$p(\tau|o_{1:H}) \propto \mathbb{I}(p(\tau) \neq 0) \prod_{t=0}^{H} p(o_t = 1|b_t, a_t). \tag{6}$$

A high probability of optimality for a given belief state and action pair should reflect a higher reward for that pair of belief state and action (as higher optimality denotes a higher reward). Optimality should then be defined so that the total probability of optimality reflects the total reward sum. To transit between this total probability of optimality product and total reward sum, the probability of optimality for a given belief state and action is set to the exponentiation of the reward of the same belief state and action:

$$p(o_t|b_t, a_t) = e^{r(b_t, a_t)} \tag{7}$$

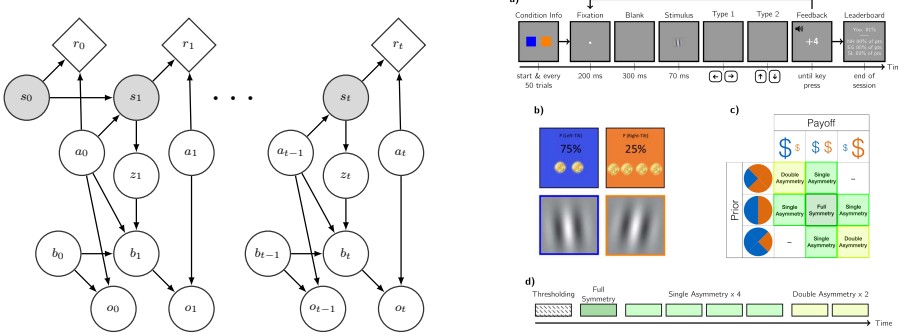

Figure 1: Left: Graphical model of the framework that measures the probability of making the optimal decision as confidence. Right: Experimental setup of a perceptual decision making task with varying prior and reward distribution (picture from [11]).

To make $p(o_t|b_t, a_t)$ between 0 and 1, we can make all rewards negative by subtracting the maximum possible reward from them.

Notably, the above choice still looks arbitrary as there are many other functions that satisfy the constraints of our problem. Beyond being an intuitive way to relate a probability and a sum, if optimality is defined for an agent as it is in equation 7, then the agent will jointly maximize the total reward and the information entropy of the policy, which is also why our framework is a *normative model*. This result can be shown by deriving the policy from the optimal trajectory as defined above. One way to derive such a policy is to approximate $p(\tau|o_{1:H})$. If the approximation of $p(o_t = 1|a_t, b_t)$ is expressed with policy $\pi(a_t|b_t)$, the approximation of optimal trajectory will be:

$$\hat{p}(\tau) = p(b_0) \prod_{t=0}^{H} p(b_{t+1}|b_t, a_t)\pi(a_t|b_t). \tag{8}$$

and the desired policy $\pi(a_t|b_t)$ can be obtained by minimizing the KL-divergence (maximizing negative KL-divergence) between the approximate and true distributions:

$$-DL(\hat{p}(\tau)||p(\tau|o_{1:T})) = \mathbb{E}_{\tau \sim \hat{p}(\tau)}\left[\sum_{t=0}^{H} r(b_t, a_t) - \log \pi(a_t|b_t)\right]$$

$$= \sum_{t=0}^{H} \mathbb{E}_{(b_t, a_t) \sim \hat{p}(b_t, a_t)}\left[r(b_t, a_t) + \mathcal{H}(\pi(a_t|b_t))\right] \tag{9}$$

This derivation is known as "planning as inference" in the literature [7, 8], and equivalent of soft Q-learning on the belief state [9, 10]. Moreover, these equations also show that our definition of the probability of optimality works even when the dynamics of the belief state are stochastic.

### 3.1 Interpretation

According to our model, the agent makes decisions strictly optimally, like a POMDP. Its evaluation of optimality, however, allows for other trajectories through the concept of soft optimality. We only derived that policy of confidence judgment to demonstrate better why exponentiation of reward (equation 7) is a reasonable choice. This self-assessment is, to some degree, similar to inverse reinforcement learning. Notably, entropy regularization has been proven to be a practical approach in inverse reinforcement learning too [9]. In this approach, the evaluator allows some suboptimality to consider noises and hidden information. Such allowance in our confidence model considers imperfect learning of the environment or possible trial-to-trial changes in the computations. For example, arousal level affects perception and, consequently, performance. While in each trial, the most rewarding action should be chosen based on the received information, the evaluation should consider variation in the quality of this information gathering, e.g., "What if this decision was made under a low arousal level?" Maximum Entropy regularization is an agnostic approach to these variations and confounding factors.

Notably, the regularization usually accompanies a parameter $\beta$ with a positive value, and $p(o_t|b_t, a_t) = e^{\beta(b_t, a_t)}$ ($\beta \geq 0$) still reflects the monotonic relationship between confidence (probability of being optimal) and the accumulated reward. We did not use that parameter in our equations for simplicity, and the fact that such an extra free parameter did not improve our fits to experimental data. However, with the extra parameter $\beta$, this model also include the hard optimality as $\beta \to \inf$.

## 3.2 Using the model on experimental data

Our framework proposes a confidence model for any sequential decision-making task under uncertainty. Specifically:

$$\text{confidence} \propto \prod_{t=1}^{H} p(o_t|b_t, a_t) = \prod_{t=1}^{H} e^{\mathbb{E}_{s_t}[r(s_t, a_t)]} \tag{10}$$

when the trajectory of the subject is $\tau = a_1, b_2, a_2, b_3, \ldots, a_H$ in a given trial. Fitting this equation to a subject's reported confidence could be extremely computationally expensive. One of the main reasons behind such computational cost is the belief state, with a space size exponential in the original state space size in the most general cases. Another reason is the need for normalization, which requires the calculation of the probability of optimality for all possible trajectories. In practice, however, fitting confidence is not intractable due to the simplicity of models of perception and the existing experimental setups. First, behavioral and even neural data on perception have been shown to be accurately modeled with Gaussian distributions. Second, in the current experimental setups, the dynamics of the task (transition function) are often very simple, and the number of actions is very minimal, e.g., one action of choosing one of the options after a perceptual cue. As a result, the belief state can always be expressed with a Gaussian distribution with two parameters of mean and variance. In other words, $b_t$ could often replaced with two one-dimensional parameters of $\mu_t$ and $\sigma_t^2$. Finally, if the task has only one step of action selection, which is often the case, the confidence is simply proportional to $e^{\mathbb{E}_s[r(s,a)]}$ where $a$ is the chosen action. While this is convenient in terms of fitting, it brings another challenge. Any monotonic function of reward as the probability of optimality aligns it with the maximum total reward concept (instead of the exponential we used in equation 7 to map summation to product). For example, one convenient heuristic is the ratio of the expected reward of different actions, i.e., confidence $\propto \mathbb{E}_s[r(s, a)]$. Although this is arbitrary, we tested this intuitive definition of confidence, called "expected value ratio", and compared it to our definition of confidence on experimental data.

## 4 Results

We tested our model in two experiments, each focusing on different aspects and potential issues of confidence modeling. The first experiment was a perceptual decision-making task with asymmetric priors and rewards [11]. We focused on the interaction of reward and perception with confidence in our fits and analyses. The second experiment was a value-based decision-making task with no perceptual ambiguity [23], in which we tested our definition of optimality and compared our results with "expected value ratio" as a confidence hypothesis. To make our results more readable, we call our confidence model, "soft optimality" decision confidence.

### 4.1 Perceptual Decision Making Task with varying priors and rewards

**Experiment:** This experiment was designed to study the interaction of reward and priors with perceptual confidence [11]. In this experiment, 10 subjects were shown a Gabor filter tilted left or right and asked to report their perception of the direction of the Gabor filter (left or right) and, subsequently, their confidence in their perception (low or high) (as seen in Fig. 1, right plot). The difficulty of the trials (the extremity of the orientation of the Gabor filters) was constant across trials and was fitted to each subject before the main experiment to have approximately 70% accuracy. Each subject completed different sessions of this task where the prior probability distribution and the reward distribution for correction choices across the two directions were varied (prior probability was either 3:1, 1:1, or 1:3; payoffs were either 2:4, 3:3, or 4:2). The subjects were told the exact prior and reward distribution before each session and every 50 trials during the session. Each session was completed on separate days, with the fully symmetric trials being completed first, followed by the asymmetric trials in a random order. Each session consisted of 700 trials, the first 100 of which were

Table 1: AIC values for the fit to all confidence models in perceptual decision-making task

| Model | 1 | 2 | 3 | 4 | 5 |
|---|---|---|---|---|---|
| Perception | $1350.81 \pm 42.17$ | $\mathbf{1668.53 \pm 2.98}$ | $1763.63 \pm 16.18$ | $\mathbf{1302.93 \pm 15.63}$ | $\mathbf{1441.89 \pm 3.19}$ |
| Soft Optimality | $\mathbf{1199.72 \pm 0.38}$ | $1819.12 \pm 20.12$ | $\mathbf{1674.94 \pm 3.02}$ | $2056.63 \pm 40.63$ | $2309.59 \pm 32.36$ |
| Observation | $1316.56 \pm 22.86$ | $\mathbf{1675.12 \pm 6.84}$ | $\mathbf{1673.77 \pm 3.03}$ | $\mathbf{1302.93 \pm 15.63}$ | $\mathbf{1441.89 \pm 3.19}$ |
| Expected Value Ratio | $1346.6 \pm 40.16$ | $\mathbf{1668.48 \pm 1.82}$ | $1695.27 \pm 7.89$ | $\mathbf{1295.91 \pm 13.51}$ | $\mathbf{1441.34 \pm 2.37}$ |

| Model | 6 | 7 | 8 | 9 | 10 |
|---|---|---|---|---|---|
| Perception | $1383.24 \pm 6.86$ | $\mathbf{1670.72 \pm 5.89}$ | $1948.82 \pm 52.65$ | $1777.82 \pm 32.36$ | $1368.61 \pm 1.29$ |
| Soft Optimality | $1827.91 \pm 22.73$ | $1889.75 \pm 20.14$ | $\mathbf{1655.49 \pm 10.5}$ | $\mathbf{1656.61 \pm 6.59}$ | $1358.02 \pm 1.07$ |
| Observation | $\mathbf{1368.84 \pm 3.38}$ | $\mathbf{1682.17 \pm 12.09}$ | $1748.28 \pm 10.26$ | $\mathbf{1673.23 \pm 18.8}$ | $1366.24 \pm 1.47$ |
| Expected Value Ratio | $\mathbf{1376.47 \pm 4.98}$ | $\mathbf{1673.96 \pm 6.71}$ | $1875.34 \pm 42.0$ | $1723.66 \pm 25.62$ | $1384.2 \pm 1.99$ |

Table 2: Rate/Probability of each reporting high confidence in each subject and each model's prediction for the trials with value asymmetry in the perceptual decision making task

| Model | 1 | 2 | 3 | 4 | 5 | 6 | 7 | 8 | 9 | 10 |
|---|---|---|---|---|---|---|---|---|---|---|
| Experiment | 80.28 | 53.16 | 49.33 | 21.96 | 28.37 | 25.37 | 46.84 | 57.65 | 42.43 | 75.87 |
| Perception | 60.78 | 48.58 | 37.17 | 16.22 | 26.93 | 17.57 | 51.13 | 29.77 | 26.34 | 68.1 |
| Soft Optimality | 79.29 | 68.72 | 52.74 | 61.96 | 69.84 | 55.78 | 68.42 | 50.17 | 47.0 | 69.5 |
| Observation | 63.35 | 55.52 | 65.06 | 16.22 | 26.93 | 23.34 | 55.99 | 43.62 | 38.82 | 68.43 |
| Expected Value Ratio | 61.06 | 49.17 | 43.4 | 16.89 | 27.42 | 18.82 | 51.76 | 32.77 | 29.69 | 83.87 |

discarded from analysis. Subjects received rewards of $0-$20 based on their performance. Notably, the subjects were explicitly instructed to report their confidence in the direction of the stimulus. In other words, they were explicitly requested to report their perception confidence. More details can be found in the original paper of this study [11].

**Fitting and Comparison:** First, we built each subject's model with a POMDP based on their choices. This POMDP contained 2 (hidden) states, each representing one of the directions. The perception of the subject (observation function) was modeled as a Gaussian distribution. Observations came from $\mathcal{N}(-1, \sigma_z^2)$ and $\mathcal{N}(1, \sigma_z^2)$ from direction left and right respectively. Notably, the actual generative process of observations could be different from the learned model by the subject [18]. Therefore, while observations were sampled from $\mathcal{N}(\pm 1, \sigma_z^2)$, the subject's internal model was $\mathcal{N}(\pm 1, \sigma_{sz}^2)$. $\sigma_z^2$ is called external observational noise. $\sigma_{sz}^2$ is the internal observational noise. Using the POMDP framework, the choice in each trial was obtained by sampling from the true generative process (with external noise), updating the belief based on the prior and learned observation function (with internal noise), and finally, picking the direction with highest expected reward by combining belief and the reward of each direction.

The external observational noise was first fit to each subject's choices in symmetric trials through gradient descent. Only in asymmetrical trials would the internal observational noise impact the subject's choice. Therefore, the internal observational noise was fit to the subject's choices in a subset of the trials with prior distribution asymmetries using a grid search and a maximum likelihood estimation with a Bernoulli likelihood function. The prior distribution was set to the actual prior value communicated to the subjects. Therefore, our model had only two free parameters in the fitting process, i.e., internal and external noise.

Based on the noise parameters obtained from the choice data, we could predict the subject's confidence in each trial according to different confidence models. The main two models were soft optimality confidence (ours) and perception confidence, which is the belief about the choice. We also included two more models. One was the observation likelihood (belief but without considering the prior), which we called observation confidence. The other model was the expected value ratio, as discussed before. For each confidence model, the confidence criterion, the threshold at which confidence is binarized into "low" or "high", was fit to the subjects' confidence reports in trials with asymmetric prior and fully symmetric trials (same subset of trials that we obtained noise parameters from). This fitting process included a grid search and a maximum likelihood estimation with a Bernoulli likelihood function.

Table 3: AIC values for the fit to all confidence models with an additional parameter accounting for the choice bias

| Model | 1 | 2 | 3 | 4 | 5 |
|---|---|---|---|---|---|
| Perception | $1364.92 \pm 32.82$ | $\mathbf{1670.7 \pm 3.61}$ | $1774.34 \pm 21.38$ | $\mathbf{1301.69 \pm 12.47}$ | $\mathbf{1443.22 \pm 4.41}$ |
| Soft Optimality | $\mathbf{1201.72 \pm 0.65}$ | $1818.99 \pm 16.09$ | $\mathbf{1676.09 \pm 4.47}$ | $2072.39 \pm 42.81$ | $2336.31 \pm 77.72$ |
| Expected Value Ratio | $1364.15 \pm 33.0$ | $1674.23 \pm 2.88$ | $\mathbf{1669.71 \pm 12.56}$ | $\mathbf{1315.26 \pm 17.07}$ | $\mathbf{1433.92 \pm 11.62}$ |

| Model | 6 | 7 | 8 | 9 | 10 |
|---|---|---|---|---|---|
| Perception | $\mathbf{1386.16 \pm 12.54}$ | $\mathbf{1699.3 \pm 31.19}$ | $1938.9 \pm 41.8$ | $1767.18 \pm 19.07$ | $1371.14 \pm 2.06$ |
| Soft Optimality | $1828.37 \pm 19.27$ | $1972.15 \pm 93.6$ | $\mathbf{1657.1 \pm 7.11}$ | $\mathbf{1659.87 \pm 5.6}$ | $\mathbf{1360.43 \pm 1.84}$ |
| Expected Value Ratio | $\mathbf{1379.42 \pm 9.61}$ | $\mathbf{1669.33 \pm 23.49}$ | $1866.07 \pm 29.56$ | $1709.11 \pm 16.48$ | $1385.87 \pm 3.81$ |

We tested the four mentioned models on the trials that had an asymmetry in the value distribution, which were not used to fit any of the parameters of these models. Notably, the model's values/rewards of actions were set to the values communicated to the subjects and were not free parameters. The AIC value ranges in table 1 are the results of fitting the model 10 times on randomly shuffled trials. The bolded values are the lowest AIC scores for a given subject, factoring in the range of uncertainty found in the average process across 10 trials. Moreover, table 2 shows the rate/probability of high-confidence reports by each subject and the prediction of the models in these trials (asymmetrical values; averaged over the 10 runs). Comparing these predictions to the actual high-confidence rate is not statistically reliable but more intuitive to humans.

As shown in table 1, our decision confidence model (soft optimality) was a better fit for five subjects compared to the perceptual confidence model. This is especially important because the subjects were explicitly instructed to report their perception confidence, yet half reported their confidence in choice. To be more precise, half could not override the assessment mechanism of their decisions. Importantly, adding the other two models (observation and expected value ratio) did not change our result. In fact, they strengthened our claim. The AIC value of our model often differs more significantly from the other three models. In other words, when our model performs better, it usually outperforms others significantly (and vice versa). This shows that the phenomenon we are observing is not an artifact of misuse of statistical tests (e.g., test's assumptions not holding) or fitting a powerful function.

We also got the same results while considering a choice bias in subjects towards one of the directions (as seen in table 3). We modeled this bias by adding a free parameter to the prior of each subject. We did not include the observation model as a comparison for these results because the observation model does not make use of a subject's prior distribution in its confidence computation. All subjects' behavior that our model fit best originally were also best explained by our model after the addition of the bias parameter.

## 4.2 Value Based Decision Making Task

To further test our soft-optimality assumption, especially compared to the expected value ratio heuristic, we applied our model to value-based decision making experiment [23].

**Experiment:** In this experiment, 33 subjects first did a rating task where they were shown pictures of different foods in succession and were asked to rate, on a sliding scale, how much they would want to eat that food at the end of the experiment (as seen in Fig. 2, left plot). After rating each food once, they were shown each food again and asked to rate it a second time to account for noise in their subjective assessment of the food's value. They were not told they would have to rate each food twice, so this didn't affect their initial rating of the food or allow them to try to memorize their first rating when rating a food for the second time. Subsequently, they were asked to do a decision-making task where they were presented with pictures of two of the foods they had previously rated and had to select which of the two they would rather eat and rate their confidence in their selection on a sliding scale. A "correct" decision was defined as a decision in which the subject chose the food object that they had rated higher on average between the two rating trials. Further task details are available in the original paper [23].

**Fitting and Comparison:** To generate predictions of subjects' confidence for the trials in this experiment, we used their average rating of a food during the rating trials to be an estimate of their

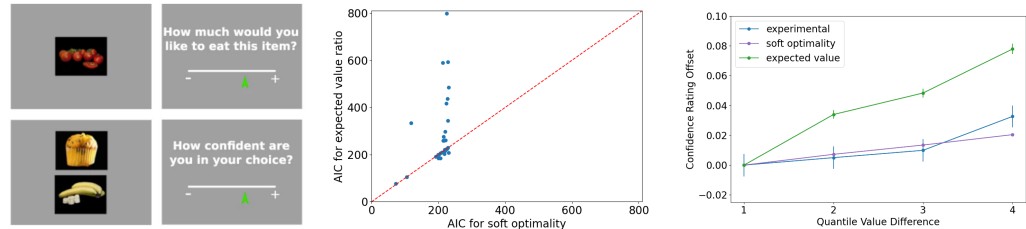

Figure 2: Left: Experimental setup of a value-based decision task with rating and decision trials (picture from [23]). Middle: Visual demonstration of AIC values for the soft optimality model versus the expected value ratio model on the value-based decision making task. The red dashed line represents equal AIC values. Right: As the quantile value difference increases, soft optimality predicts a rate of confidence increase closer to the experimental rate as compared to the rate predicted by the expected value method.

Table 4: AIC values by subject for different confidence prediction models on the value-based decision making task

| Model | 1 | 2 | 3 | 4 | 5 | 6 | 7 | 8 | 9 | 10 | 11 |
|---|---|---|---|---|---|---|---|---|---|---|---|
| Expected Value Ratio | 258.87 | **224.61** | **219.06** | 589.85 | **210.31** | 333.47 | 206.60 | **202.14** | **190.56** | 799.15 | 219.69 |
| Soft Optimality | **216.79** | 225.63 | 220.58 | **212.84** | 210.84 | **118.28** | **206.40** | 218.32 | 191.94 | **224.71** | **219.48** |

| Model | 12 | 13 | 14 | 15 | 16 | 17 | 18 | 19 | 20 | 21 | 22 |
|---|---|---|---|---|---|---|---|---|---|---|---|
| Expected Value Ratio | **185.37** | 261.71 | 297.92 | 220.79 | 223.33 | **215.82** | 260.02 | 417.31 | **199.57** | 229.26 | 199.56 |
| Soft Optimality | 200.20 | **221.03** | **219.84** | **219.07** | **223.21** | 218.58 | **214.67** | **223.19** | 200.84 | **228.74** | **199.30** |

| Model | 23 | 24 | 25 | 26 | 27 | 28 | 29 | 30 | 31 | 32 | 33 |
|---|---|---|---|---|---|---|---|---|---|---|---|
| Expected Value Ratio | **207.01** | **215.36** | **104.43** | 343.86 | 220.60 | **185.43** | 484.97 | 436.96 | 276.38 | 76.54 | 593.16 |
| Soft Optimality | 230.68 | 219.77 | 105.06 | **228.51** | **220.16** | 206.52 | **230.81** | **227.34** | **215.55** | **71.99** | **228.66** |

expected value of reward for a given option during the choice tasks. We then used this expected value of the belief state (where the belief state is their hypothesis as to which of the two choices is the "correct" choice) in our soft optimality equation and normalized this value to map our confidence prediction to the sliding scale that subjects use to report their confidence in their decision.

To assess our model's ability to predict subjects' confidence reporting behavior, we compared the fit of our soft optimality model to the fit of the expected value ratio model (as seen in Table 4 and Fig. 2, middle plot). Notably, since we used the values obtained from the rating section, none of the models have any free parameters. The additional two models used for comparison in the perceptual decision-making experiment couldn't be used as metrics of comparison in this task as there was no perceptual component, which is a requirement of using the observation and perception models. On average, our soft optimality outperformed the expected value model with an average AIC value of 206.65 across all subjects as compared to an average AIC value of 282.11. The soft optimality model was also a better fit for the majority of the subjects; it best explained 21 out of the 33 subjects' confidence reporting behavior. Moreover, the difference in AIC values is significantly larger in subjects whose behavior is better explained by our model (soft optimality) as demonstrated in Fig. 2, middle plot. We also compared the relationship between the difference in the value of the offered choices and the subject's confidence (relative to their minimum confidence level, i.e. confidence offset, for normalization across all subjects) with the predictions of the two models. Our method better predicts the rate at which the subjects' confidence will correspondingly increase (Fig. 2, right plot). Validating our model on this data set strengthens our conviction that it can generalize across multiple types of common decision paradigms and that it can be descriptive of more subjects than other comparable Bayesian approaches. Moreover, it further confirms the practicality of the exponential function in explaining confidence as opposed to heuristics like the expected value ratio.

# 5 Discussion

We present a normative framework that measures "decision confidence" and captures its interaction with reward, perception, and prior. Our model is essentially planning as inference framework, developed and discussed before [7, 10]. Our main contribution here is connecting this framework to the confidence judgment in humans. Notably, soft reinforcement learning has been used to fit subjects' choices in decision-making before. However, those were actual decisions in the context of exploration-exploitation or imperfect rationality. We used this approach in action evaluation. In fact, according to our model, actions themselves follow strict optimality (modeled by a POMDP). We validated our results on two different experiments from different groups. Both datasets are publicly available and accessible in their corresponding papers [11, 23]. Our analysis code is also available at https://github.com/ameliamj/decision-confidence-model.

Current experiments, mostly including only 2 choices and 1 step of action selection, are insufficient to flesh out different aspects of confidence and its models. We believe the generalizability of our framework makes it a great candidate for testing and modeling confidence in more complicated setups. Future experiments could go in multiple directions. One example is confidence assessment in a sequence of actions instead of one action. One important aspect of our model, which is not present in other models and even most experiments, is the ability to assess confidence in a sequence of actions (trajectory). Multiple common experimental setups in the field, such as two-step task for humans [24] and even maze navigation for rodents can be used to study confidence of a sequence of actions (whole decision) if confidence assessment is added to them. Another example direction is confidence assessment in multiple choices. Two-choice tasks are too simple to distinguish between various models. Therefore, some researchers have started focusing on confidence assessment on multiple choices. For example, one study has shown that the difference between the probability of the top two choices explains confidence better than the posterior probability of the most likely state when three choices are presented [5]. In that study, the reward was the same for all options. It could also be illuminating to test confidence assessment when the rewards of different choices are not equal.

Confidence expression is an important factor in society and affects individuals' interactions. Underprivileged individuals often suffer from low confidence, which may further hurt them in their role in their community. Therefore, we believe accurate estimation of confidence and comparing it with actual performance will reduce the societal gaps and promote fairness. However, computational models of confidence and, in general human behavior are very recent. Therefore, no impactful decision should be made based on these models (e.g., in job applications) until they are tested extensively in numerous controlled setups.

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
