# OpenReview forum: "A Unifying Normative Framework of Decision Confidence"
_NeurIPS.cc/2024/Conference — NeurIPS 2024 poster_

### Official Review · Reviewer_R2CF · 2024-07-04

**Soundness:** 4
**Presentation:** 3
**Contribution:** 2
**Rating:** 6
**Confidence:** 4

**Summary:**

Great paper about confidence with payoffs but maybe not the unifying one yet. They follow the Bayesian Confidence Hypothesis and the optimality assumption to formulate in a POMDP the modelling of perceptual confidence and decision making. The title should be more like modeling decision confidence as soft Q-learning under the reward optimality assumption.

**Strengths:**

1.	The motivation is clear and has impact in the decision making community
2.	The example is illustrative.
3.	The mathematical formalism is sound and the Bayesian Confidence Hypothesis is well known.
4.	The evaluation with real human data.
5.	Replicability with databases available and code.

**Weaknesses:**

1.	Assimilating rewards to different confidence is only valid under the RL framework. In essence is the task that has different confidence.
2.	We model decision confidence as “probability of making the best decision”. This is a strong assumption that drives the whole work. But it may be not the only description. Thus, going against the unified framework. See Fleming 2019. Self-Evaluation of Decision-Making: A General Bayesian Framework for Metacognitive Computation. And its follow-up statement: “decision confidence is the probability of being optimal over a sequence of states and actions given the policy.” It may be in terms of the mathematical conceptualization of the authors, but what about the posterior entropy of the policy? Or the posterior uncertainty of the state-estimation? In the models, this is better clarified: “their confidence in their choice closely matches the probability of choosing the correct option, i.e., the posterior probability of the most probable choice”. I think optimality is a rational agent biased concept. Actually, the issues of optimality are mentioned in the problem definition. Equation 10 is then a strong assumption.
3.	The approach may fail to go beyond more than binary forced choice scenarios. i.e. intuitively, we would expect our confidence in a selected action in a 3 choice scenario to be lower if another unchosen option was extremely close in probability.
4.	In the results, the paper could be more interesting if comparing with posterior entropy of the belief state, the action model and the observation model.

**Questions:**

Does planning as inference framework always incorporate the information entropy?
The summary on MDP, POMDP and belief-MDP is nice but is it actually needed?
An example of the latter case is Kalman-filter-like environments where the belief state can always be represented with a Gaussian distribution with two parameters of μt and σt. Not only in Kalman, we can always model the belief state into a variational (factorized) density formed with Gaussians. Furthermore, we can use an encoder to reduce dimensionality. This is the SOTA approach to model-based RL.
“optimal agent such as POMDP” is a POMDP an optimal agent or a mathematical formalism to model decision making.

Why defining a trajectory à la optimal planning as inference if every trial is independent?
While it is presented as that the derivation matches the planning as inference, it may be the other way around. So the authors forced the assumptions to end into soft Q-learning.
“This self-assessment is, to some degree, similar to inverse reinforcement learning.” Is this not too biased to problems with rewards?
Would be great to explain a mathematical intuition of perceptual confidence.
AIC tables better explanation on what would be a perfect fit could improve the clarity.

Literature suggestions:

Fleming 2019. Self-Evaluation of Decision-Making: A General Bayesian Framework for Metacognitive Computation.
Meera, A. A., & Lanillos, P. (2024). Confidence-Aware Decision-Making and Control for Tool Selection. arXiv preprint arXiv:2403.03808.


Typo: decision-make,r

**Limitations:**

Limitations are missing.

---

> ### Author Rebuttal · Authors · 2024-08-07
>
> We thank the reviewer for their constructive feedback and literature suggestions. We will incorporate them into our paper.
>
> We agree entirely that confidence could be different when more than 2 choices are available. In fact, there is a recent work showing that confidence is better modeled by the difference in probability of the top 2 choices as opposed to the posterior belief of the top choice, when three options are presented (Li 2020). Notably, we predict that this is very well modeled by the SoftMax function, where the probability of the least possible option converges to zero.  Nonetheless, testing the model on such an experiment would be a very interesting follow-up work on our model.
>
> We also fit the posterior entropy of the belief to subjects’ behavior in experiment 4.1. All AIC values were higher than those of other models (Table 1, rebuttal pdf). This is consistent with this model's poor performance on the three-choice dataset explained above (Li 2020).
>
> Regarding modeling with Markov Processes, we agree that other frameworks are also available. However, the core concepts of all of these models are the same: Markovian system and Bayesian Inference.
>
> Reference:
>
> -"Confidence reports in decision-making with multiple alternatives violate the Bayesian confidence hypothesis," Hsin-Hung Li & Wei Ji Ma, Nature Communication 2020

---

> > ### Comment · Reviewer_R2CF · 2024-08-13
> > **Good model but maybe not unifying nor normative yet**
> >
> > Thanks so much for the reference. The reading of this contribution was really inspiring, but the model is not unifying and normative yet. I would like to maintain my positive scoring. And I think it contribute more than I rated in the first evaluation. It may be a good proposal for a model but there is need for more evidence. I think the authors did a very good job using human data. Still the data and the conclusions may be confusing.
> >
> > Finally, I do not agree with the reviewer that directly rejects this paper. I think we should be fair and give a chance to this work.

---

### Official Review · Reviewer_rZps · 2024-07-10

**Soundness:** 3
**Presentation:** 4
**Contribution:** 2
**Rating:** 6
**Confidence:** 4

**Summary:**

In this work, the Authors propose a normative model of decision confidence extending the planning-as-inference framework with an optimality variable proportional to the softmax over the reward distribution. The model is fitted to the data from two behavioral tasks featuring the participants’ confidence reports; comparisons with other models of confidence are provided.

**Strengths:**

The text is written remarkably well, clear, and easy to follow.

The problem is well-motivated; the model is novel and, at the same time, rooted in literature.

The separation between perceptual confidence and decision confidence is principled, well-articulated, and put to a meaningful test through two experimental scenarios meant to distinguish the two.

The statistics are done in a robust, appropriate way.

**Weaknesses:**

In the value-based decision-making task, the proposed model best explains the confidence reports only for a fraction of the participants, although a large one. In the perceptual decision-making task, where the perceptual model was naturally expected to dominate, the proposed model best explains half of the participants’ reports. While these results are interesting in their own right, they call for a deeper dive into the issue.

An ultimate test for the model could be to look into neural activity, which naturally calls for looking into animal confidence data. Other published studies may offer such data and insights as to where to look for corresponding representations in the brain. Perhaps, a line of experimental + theoretical work by Paul Masset, Torben Ott, and Adam Kepecs (in various order/combinations) may be used to further distinguish the perceptual vs. decision confidence and to seek their representations in the brain. Either way, it would be nice to propose an experiment that would allow us to solidify the results in this paper, and derive testable predictions for such an experiment.

**Questions:**

-Please discuss the relevance (or the lack thereof) of existing behavioral/neuronal data in animals to the validation of the proposed model. Additional analyses, if possible, may strengthen the results.

-What experiment could further support the relevance of the proposed model to the confidence computation in humans/animals? What could be the additional steps to verify the proposed model?

-Please discuss your results on why different models are best at explaining the confidence reports of different participants. Is it because different participants use different confidence models or is it because there might be a third model that universally explains their reports? What does the literature have to say about such scenarios?

**Limitations:**

Different models are best at explaining the data from different participants.

---

> ### Author Rebuttal · Authors · 2024-08-07
>
> We thank the reviewer for bringing up great points about testing the model in more and better experiments. We agree that these directions need to be discussed in the paper, and we will do so in the final version, which allows one more page.
> Current experiments, mostly including only 2 choices and 1 step of action selection, are insufficient to flesh out different aspects of confidence and its models. We believe the generalizability of our framework makes it a great candidate for testing and modeling confidence in more complicated setups. Future experiments could go in multiple directions. Here are a couple of examples:
> 1. Confidence assessment in a sequence of actions instead of one action: One important aspect of our model, which is not present in other models and even most experiments, is the ability to assess confidence in a sequence of actions (trajectory). Multiple common experimental setups in the field can be used to study confidence of a sequence of actions (whole decision) if confidence assessment is added to them:
>
> 1.1 Two-step task: One potentially interesting experiment is assessing the confidence of humans in the two-step task (Daw 2011). The two-step task is a well-known experiment in neuroscience that involves two binary choices in each trial. However, confidence has not been assessed in that experimental setup yet. It is interesting to see how the involvement of multiple actions for each decision affects confidence and whether such an effect is compatible with our model’s prediction.
>
> 1.2 Maze navigation with perceptual cues (odor navigation): In one of the classic papers on confidence in neuroscience, Kepecs et al. used the waiting time of the mice for the “expected incoming reward” as a measure of their confidence (Kepecs 2008). Combined with the odor navigation task, one can assess the confidence of mice when multiple actions are involved (e.g. navigating the maze and sampling). Studying the role of different regions, such as the orbitofrontal cortex and hippocampus, in this confidence assessment could be an exciting area of research, too.
>
> 2. Confidence in multiple choices (with different values): As also pointed out by another reviewer, two-choice tasks are too simple to distinguish between various models. Therefore, some researchers have started focusing on confidence assessment on multiple choices. For example, one study has shown that the difference between the probability of the top two choices explains confidence better than the posterior probability of the most likely state when three choices are presented (Li 2020). In that study, the reward was the same for all options. It is interesting to see how that works when the rewards of different choices are different. Notably, the softmax function can model the difference between the two best choices, by eliminating the least possible option. Therefore, we expect that such an experiment confirms our framework’s predictions.
>
> Regarding additional results, we added another analysis in experiment 4.2, looking at the confidence rating based on the choice value difference (figure 1 one rebuttal pdf). While the confidence increases with the increase in the choice value difference in experimental data and both tested models, the predicted growth rate in our model is significantly closer to the experimental data. Specifically, our model predicts very slow growth of confidence with the value difference increase, just like the experimental data.
>
> Regarding different strategies vs. one for confidence assessment, the literature points out both individual differences (Navajas 2017) and parameters that affect confidence, such as attention, confirmation bias, and various types of noises (sensory, motor, and decision) (Khalvati 2019, Li 2020). Notably, Bayesian frameworks could incorporate these effects, but the model would have more parameters. With the increased parameters affecting decision-making and confidence, one could expect even more significant differences between subjects’ behaviors.
>
> References:
>
> -"Neural correlates, computation and behavioral impact of decision confidence,"  Adam Kepecs, Naoshige Uchida, Hatim A. Zariwala & Zachary F. Mainen, Nature 2008
>
> -"Model-based influences on humans’ choices and striatal prediction errors," Nathaniel D. Daw, Samuel J. Gershman, Ben Seymour, Peter Dayan, and Raymond J. Dolan, Neuron 2011
>
> -"The idiosyncratic nature of confidence, " Joaquin Navajas, Chandni Hindocha, Hebah Foda, Mehdi Keramati, Peter E Latham 5, Bahador Bahrami, Nature Human Behavior 2017
>
> -"Bayesian inference with incomplete knowledge explains perceptual confidence and its deviations from accuracy", Koosha Khalvati, Roozbeh Kiani, Rajesh P.N Rao, Nature Communication 2019
>
> -"Confidence reports in decision-making with multiple alternatives violate the Bayesian confidence hypothesis," Hsin-Hung Li & Wei Ji Ma, Nature Communication 2020

---

> > ### Comment · Reviewer_rZps · 2024-08-08
> >
> > Thanks for your informative response.
> >
> > I see the Authors' point that existing decision data seems insufficient to distinguish between the different models of confidence.
> > It is exciting at the same time to see the extensions of existing experimental setups that the Authors have in mind to further approach that distinction.
> >
> > As nothing more can be done just yet, I maintain my positive valuation of this work and look forward to seeing what future experimental data tells us about the mechanisms of confidence.

---

### Official Review · Reviewer_Q2rg · 2024-07-11

**Soundness:** 1
**Presentation:** 2
**Contribution:** 3
**Rating:** 3
**Confidence:** 3

**Summary:**

The present paper presents a normative framework for modeling decision confidence in humans that is generalizable to various tasks and experimental setups. In particular, the authors connect the planning as an inference framework to decision confidence. They validate their model in two different psychophysics experiments, where it is compared to other approaches in explaining subjects’ confidence reports.

**Strengths:**

In general, I believe that this work has many of the ingredients of a solid project. Linking planning as an inference to decision confidence has potential, and the goal to unify different notions of confidence is ambitious. The selected experiments allow to highlight the advantages of the proposed method.

**Weaknesses:**

1. Decision confidence is -- to a large extent -- about epistemic uncertainty, as also mentioned by the authors in their discussion of the Bayesian confidence hypothesis. Yet, any representation of epistemic uncertainty is absent in the framework at present.

2. The authors argue that their framework provides normative justifications. However, Equation 7 is never motivated from a normative perspective but rather introduced in a heuristic manner. There could be a potential for a normative justification I believe but it is never formalized.

3. The results presented in Tables 1 and 3 are rather mixed and do not show clear benefits of the proposed framework.

Furthermore, the paper would benefit from further finetuning in terms of writing. To give a few examples:

4. The authors write that "One of the applications of POMDPs and similar Bayesian frameworks is modeling the behavior “perceptual decision making” [...]". This implies that their framework is Bayesian, which it is not (see point 1).

5. The authors write that "we include the idea of optimality to the POMDP framework". This sounds like the authors invented optimality in the context of POMDPs, which is obviously not the case.

6. The optimality variable is sometimes denoted with O, sometimes with o_{1:H}. There could be more consistency.

7. "This is not ideal, especially when the agents are humans, which are inherently sub-optimal" => sounds strange.

8. Equation 5 is introduced as "the probability of a trajectory being optimal" but it rather is the probability of producing a trajectory, given that you are optimal. In general, Equation 5 could be untangled a bit more.

9. End of page 5: beta is not introduced and there is a typo in the equation. Furthermore, it should be discussed that changing beta corresponds to scaling of the reward. Perhaps this could also be a footnote if beta is not used.

**Questions:**

1. The authors write that "confidence is mainly mathematically defined only for scenarios where different choices have the same potential reward". That is not clear to me. Why is this true?

2. What is meant by asymmetric and symmetric reward functions?

3. The authors write that "According to our model, the agent makes decisions strictly optimally, like a POMDP. Its evaluation of optimality, however, allows for other trajectories through the concept of soft optimality". It is unclear to me why this distinction is necessary. If I understand it correctly, the acting part is never evaluated/considered in the experiments.

4. It was unclear to me how priors come into the model (i.e., for the first experiment).

5. How are observation likelihood and perception confidence different?

6. How is the fitting to subjects' reported confidence done? Were there any free parameters? If so, are these fitted on choices, and then evaluated on the confidences, or are they directedly fitted on confidences?

7. Is the POMDP formulation actually needed for experiment 2?

**Limitations:**

Not really discussed.

---

> ### Author Rebuttal · Authors · 2024-08-07
>
> We thank the reviewer for their feedback and questions.
>
> Our framework is normative because the agent maximizes the reward and information entropy of the policy (eq 9). We presented our approach somewhat backward to make it more intuitive for confidence representations. As other reviewers also noted, we should emphasize in the paper that eq 7 seems arbitrary and heuristic until we reach eq 9. Importantly, maximizing entropy in the fitting is actually mainly related to epistemic uncertainty, where the agent does not know the underlying model and makes minimal assumptions about it.
>
> In response to your questions:
>
> 1- Our intention with this sentence was just to communicate that the standard in the field is that confidence is mainly mathematically defined only for scenarios where different choices have the same potential reward. This is likely true because most perceptual decision-making paradigms (which is most frequently where in the past confidence has been defined) feature two choices which will give the exact same reward if they are the correct choice, given the perceptual stimulus, for the trial. We point this out because the goal of our model is to generalize to paradigms which have asymmetric distributions of reward which are much more closely related to real life scenarios.
>
> 2- A symmetric reward function is meant to denote the situation where any choice will give the same reward if it is the correct choice while an asymmetric reward function means that is the subject is correct, the reward they will receive depends on the choice they picked. As an example, the perceptual decision-making task in Section 4.1 has trials with both symmetric and asymmetric reward functions. The subject must choose if they think the stimulus was tilted to the right or left based on a brief picture of it they saw; in symmetric trials, they will get a reward of 1 if they correctly choose the direction the stimulus was tilted, while in asymmetric trials, they could get a reward of 3 if they choose right and that is the direction of the stimulus they were shown versus only getting a reward of 1 if they choose left and that is the direction of the stimulus they were shown. In this case, the asymmetric reward would encourage someone who was more unsure of their perception to choose the direction with the higher reward because that might be the option with the greater expected reward.
>
> 3- We evaluate the acting part to fit some of our model's parameters, specifically the internal and external observational noise. There is more information in lines 255-269 where we talk about fitting the subject's choices (to fit these subject’s choices we had to evaluate the acting part).
>
> 4- If you refer to Equation 10, this is the equation that we use for confidence in the later sections on experimental data; the priors from the first experiment are used to calculate the expected reward (in the exponent) of this equation. For the first experiment the equation we used for expected reward was $E_{s}[r(s,a)]= \int p (s│z)r(s,a)ds$ where $p(s│z)$  is the posterior probability of the state given the observation and so is defined as $p(s│z) \propto p(s) p(z|s)$ where $p(s)$ is the prior.
>
> 5- Observation likelihood doesn’t not include the prior information given to the subject about the trial while perception confidence does. So observation likelihood is $p(z|s)$ which is just the likelihood of the observation for that trial given the state the subject choose while perception confidence is $p(s)p(z|s)$ which combines the likelihood of the observation with the given prior information (which results in a Bayesian posterior, so this is the Bayesian confidence hypothesis).
>
> 6- We didn’t fit any parameters for experiment 2, however, we did fit some for experiment 1. The details of our fitting methods can be found in lines 248-275. We fit 3 different parameters: internal observational noise, external observation noise, and the confidence cutoff (which we used to binarize the continuous values of confidence we generated to the “high” or “low” assessments of confidence that subjects gave). External observational noise was fit using gradient descent while the other two parameters were fit using grid search with maximum likelihood estimation. Internal and external observational noise were fit on subjects’ choices while confidence cutoffs were fit on subjects’ confidence assessments.
>
> 7- The POMPD formulation is not strictly “necessary” for experiment 2 as the subject are tasked with only making one decision as opposed to the string of decisions represented in the POMDP framework. Our goal was to define a model that could generalize across many decision-making paradigms so even if the complexity is not fully utilized in experiment 2, it is included as a general framework for modeling decision-making tasks that are more complex than experiment 2.

---

> > ### Comment · Reviewer_Q2rg · 2024-08-11
> >
> > Thank you for the responses. I have decided to keep my original score and encourage the authors to submit a revised version to a future conference once the weaknesses are addressed.

---

### Official Review · Reviewer_GT57 · 2024-07-13

**Soundness:** 3
**Presentation:** 4
**Contribution:** 3
**Rating:** 6
**Confidence:** 4

**Summary:**

The submission proposes a model of decision confidence that is applicable to value-based as well as perceptual decision making tasks, and is generally grounded in normative inference. The approach is applied to two real-world datasets.

**Strengths:**

I think the motivation is compelling, grounding confidence out normatively is sensible, and a lot of detail is provided about the empirical evaluation, which is somewhat mixed but remains favorable to the proposed model in the value-based decision making dataset. I also think the introduction and background is easy to understand and exhaustively presented.

**Weaknesses:**

My two biggest concerns are [a] that the results don't favor the proposed model all that strongly or consistently, and [b] that the transformation of reward sum to probability of optimality seems no less heuristic than the alternatives.

Regarding the latter: for all that the paper makes strong claims about deriving a normative notion of confidence from first principles, expression 7 seems a bit heuristic, I'm not clear on why this definition should hold (except if it is there to back into soft Q-learning later in the section -- if so, maybe the paper should just be up front about the fact that it's taking its inspiration from those set of approaches). I'm also a bit surprised that this indicator function is used instead of something more conventional like regret (which would sidestep the issue regarding penalizing all non-optimal trajectories the same as identified on the paragraph starting line 150, and not require this extra heuristic).

Regarding the former: unless I'm missing something, the results in section 4.1 are not particularly conclusive w.r.t. which model best explains the data. I'm partial to the idea that different participants interpret the confidence prompt differently (cf. Ericsson and Simon 1984) but I'm not convinced by the argument that this is actually a result in favor of the proposed model because participants can't "override their intrinsic mechanism". The results in section 4.2 are a bit more conclusive, though it can be hard to interpret AIC (vs something like out of sample predictions, or even cross-validated log likelihoods). I'm wondering whether we're seeing an artifact of using an exponential instead of a ratio, where it's easier for the exponential model to saturate and predict very high / low confidences, yielding the relatively large AIC values.

**Questions:**

* L321-322 I cannot parse "we compared the fit of our soft optimality model to the expected value ratio model explained away". Can you reword?
* I'm confused about the definition of probability of optimality. I would imagine that there's a function o:f(tau)->[0,1] that maps trajectories to optimal or not, which then induces a probability distribution P(o=1|a..., b...) because of the stochasticity in trajectories. Why is the conditional flipped instead on line 148?
* L132 typo "decision-make,r"

**Limitations:**

Adequately discussed.

---

> ### Author Rebuttal · Authors · 2024-08-07
>
> We appreciate the reviewer's constructive feedback and comments.
>
> Concerns:
> About the results, we would like to emphasize that in the first experiment (4.1), subjects were explicitly instructed to report their perceptual confidence (lines 245-247). However, half of them reported their decision confidence (line 287). That’s why we talked about overriding. Notably, even the observation of different effects of prior and value on the subject’s confidence was important and interesting enough that the original paper was published in a prestigious venue in the field of psychophysics a few years ago. That paper was only about the different effects of prior and payoff on various subjects, and it didn’t talk about decision confidence nor a normative model for that. Our model explains that observation in a systematic way. In other words, there was no normative model for that subset of subjects, and we propose one in this paper.
>
> For the second experiment, we added another analysis. This result provided more evidence in favor of our model. As demonstrated in Figure 1 of the rebuttal pdf, we looked at how confidence rating changes based on the difference in the value of presented choices. While the confidence increases with the increase in the choice value difference in experimental data and both tested models, the predicted growth rate in our model is significantly closer to the experimental data. Specifically, our model predicts very slow growth of confidence with the value difference increase, just like the experimental data.
>
> Regarding the exponential function, which looks like a heuristic by itself, we agree with the reviewer that our framework is based on the soft-Q learning principle, especially as a normative model. We chose this narrative as we found it more intuitive for the concept of confidence. We will add more explanation upfront about expression 7 and emphasize that it seems arbitrary until we reach the soft-Q learning loss function.
>
> Questions:
> 1-Sorry for the typos; The sentence should be “To assess our model’s ability to predict subjects’ confidence reporting behavior, we compared the fit of our soft optimality model to the fit of expected value ratio model.”
>
> 2- The reviewer is right; line 148 should be replaced with “Consequently, the probability of observing trajectory τ in an optimal agent is p(τ |O = 1, b1), which should be maximum for a trajectory that is generated by the optimal policy π∗".

---

> > ### Comment · Reviewer_GT57 · 2024-08-13
> >
> > Regarding perception vs decision confidence -- I understand the argument (that participants were instructed to report their perceptual confidence, but according to the model a subset reported their decision confidence instead). But we can't both use the model to make this interpretation, *and* also use this interpretation in support of the model being good (that appears to be a circular argument).
> >
> > I do appreciate the additional analysis showing another "signature phenomenon" match to the proposed model, and I do think that foreshadowing the connection to soft Q-learning feels more honest, but the choice of that heuristic doesn't share the strength of the normative argument.
> >
> > Based on the above points, I feel comfortable keeping my score as-is.

---

### Author Rebuttal · Authors · 2024-08-07

Content of PDF: Plot of additional analysis for experiment 2. AIC table of Entropy model for experiment 1.

---

### Decision · Program_Chairs · 2024-09-25

**Decision:**

Accept (poster)

**Comment:**

Modeling subjective decision confidence is a popular topic in the cognitive sciences. From a psychological perspective, understanding how humans self-assess their behavior is of fundamental importance. The work describes and validates a new model formulation for decision confidence based on a POMDP model framework suitable for describing decision behavior in value-based and perceptual
decision tasks.

The strength of the paper is that in contrast to the vast majority of existing confidence models, it present a formulation for decision-confidence that takes into account the reward structure of the task. The model considers subject's confidence reports as indicating their beliefs in having made the "best" decision while defining "best" in terms of highest expected reward. The confidence formulation is normative within the POMDP framework assuming the decision objective is to jointly maximize expected reward and information entropy of the decision policy. Validation against data from multiple human confidence experiments support the model. Also during the rebuttal, the authors presented model fits/predictions for confidence ratings in a three-alternatives perceptual decision task, showing the generalizability of their model.

Reviewers raised minor concerns with regard to the claim of normativity and the model performance in one of the tested datasets. The rebuttal mostly resolved these issues although the formulation remains limited to the assumed optimal behavioral model. However, the paper is well written and describes work that is conceptually novel, technically solid, and has the potential to significantly impact the field of studying human decision confidence.